# Facile Synthesis and Electrochemical Characterization of Polyaniline@TiO_2_-CuO Ternary Composite as Electrodes for Supercapacitor Applications

**DOI:** 10.3390/polym14214562

**Published:** 2022-10-27

**Authors:** Nadia Boutaleb, Fatima Zohra Dahou, Halima Djelad, Lilia Sabantina, Imane Moulefera, Abdelghani Benyoucef

**Affiliations:** 1Department of Chemistry, Faculty of Science, University of Dr. Moulay Tahar, Saida 20000, Algeria; 2Physics Department, Faculty of Exact Sciences, University of Mustapha Stambouli, Mascara 29000, Algeria; 3Department of Physical Chemistry, Institute of Materials, University of Alicante, 03080 Alicante, Spain; 4Junior Research Group “Nanomaterials”, Faculty of Engineering and Mathematics, Bielefeld University of Applied Sciences, 33619 Bielefeld, Germany; 5Chemical Engineering Department, Campus Universitario de Espinardo, University of Murcia, 30100 Murcia, Spain; 6L.S.T.E. Laboratory, Faculty of science and technology, University of Mustapha Stambouli, Mascara 29000, Algeria

**Keywords:** copper(II) oxide, titanium dioxide, polyaniline, ternary-composite, supercapacitors

## Abstract

This research reports the facile, controlled, low-cost fabrication, and evaluation of properties of polyaniline matrix deposited on titanium dioxide and copper(II) oxide ternary-composite (PANI@TiO_2_–CuO)-based electrode material for supercapacitor application. The process involves the preparation of CuO in the presence of TiO_2_ to form TiO_2_–CuO by a facile method, followed by in-situ oxidative polymerization of aniline monomer. The structural and physical properties were evaluated based on the results of FTIR spectroscopy, X-ray diffraction (XRD) analysis, X-ray photoelectron spectroscopy (XPS), transmission electron (TEM) and scanning electron (SEM) microscopy, thermogravimetric analysis (TGA), and BET surface areas analysis. The results indicated that TiO_2_–CuO was dispersed uniformly in the PANI matrix. Owing to such dispersion of TiO_2_–CuO, the PANI@TiO_2_–CuO material exhibits dramatic improvements on thermal stability in comparison with the pure PANI. The cyclic voltammetry (CV) confirms the reversibility of PANI redox transitions for this optimized electrode material. Moreover, the results reveal that the specific capacitance of PANI@TiO_2_–CuO reaches 87.5% retention after 1500 cycles under 1.0 A g^−1^, with a better charge storage performance as compared to pure PANI and PANI@TiO_2_ electrodes. The preparation of PANI@TiO_2_–CuO with enhanced electrochemical properties provides a feasible route for promoting its applications in supercapacitors.

## 1. Introduction

Recently, hybrid material composites have been extensively studied, owing to their potential utilities. They contain the unique properties of nanofillers, driving the materials with improved properties [1,2,3]. Moreover, conductive polymers (CPs) and their composites receive considerable attention due to their magnetic, optical, and opto-electronic characteristics [4,5]. Polyaniline (PANI) is one of the most common conducting polymers due to its distinct environmental stability, superior chemical/physical properties, thermal stability, ease to synthesis, and reversible doping/dedoping process [6,7,8]. They are commercially important polymers used extensively in many applications [9,10]. In recent years, hybrid materials based on CPs and inorganic materials have been investigated at length by incorporating the inorganic materials in a CP matrix. Furthermore, the final properties of nanocomposites depend not only on the properties of the individual components, but also on the homogeneity of the dispersibility of the inorganics, and the interfacial between the interacted components [11,12]. On the other hand, to improve environmental stability, the metal oxide (MO) nanoparticles were added into CPs. The development of established technology is a result of the hybridization of nanocomposite to build an outstanding nanomaterial with specific features that outweigh the sum of their individual contributions [13]; because of their higher degradation efficiency, good physical and chemical properties, and low toxicity, CPs@MO nanomaterials have fascinated researchers for their excellent properties [13,14,15]. Among MOs, Titania (TiO_2_) is a promising product because it is abundantly available in nature and is environmentally friendly [15,16,17]. As a suitable potential candidate, it has already been authorized in a broad range of different applications such as sensors, photocatalysis, solar cell, and anticorrosion [18]. Researchers have proclaimed the structural, optical, and electrical properties of PANI-based TiO_2_ nanomaterials as an electrode material for supercapacitors, such as Deshmukh et al. [19] proved that the distinctive structure of the PANI–TiO_2_ composite and the cohabitation of conducting PANI with TiO_2_ have been found to be responsible for the superior electrochemical properties. Ghahramani et al. [20] synthesized PANI-coated reduced graphene oxide for removal of malachite green from aqueous solution. According to the results of Yoruk et. al. [21], the rGO/TiO_2_/PANI nanocomposite shows much better performance than its individual components in terms of specific capacitance, energy density, power density, and cycle life. Nevertheless, copper oxide (CuO) is also a p-type semi-conductor and has a narrow band gap of 1.2–1.9 eV, as well as high surface areas and unusual morphology structures, which provides it with characteristic physical and chemical properties that have been utilized in a number of applications such as magnetic storage, energy storage, photocatalysis, energy conversions, superconductors, solar cells, antimicrobial agents, and gas sensors [22,23]. Moreover, PANI, TiO_2,_ and CuO can supplement each other and their synergistic effect can improve the performance of the hybrid material of PANI@TiO_2_-CuO. For instance, Nekooie et al. fabricated a CuO/TiO_2_/PANI to obtain a composite for chlorpyrifos depollution in water by visible light irradiation [24]. Pang et al.’s TiO_2_-SiO_2_ and CuO-TiO2-SiO2 nanocomposite fibers were formed by electrospinning with subsequent calcination. Then, in-situ chemical oxidative polymerization of aniline monomer was carried out with inorganic (TiO_2_-SiO_2_ and CuO-TiO_2_-SiO_2_) nanocomposite fibers as templates [25].

Presently, the ternary nanocomposite is becoming an emerging area of research. In our present work, we have prepared PANI@TiO_2_–CuO via an in-situ chemical oxidative polymerization method. Relatively cheap and less hazardous raw materials are used for the synthesis, and thus make this process both economic and eco-friendly. The interaction phenomenon between PANI and CuO with TiO_2_ was proven by XPS, FTIR, XRD, UV-vis, BET, TGA, TEM, and SEM techniques. Moreover, cyclic voltammetry (CV) was applied to study the electrochemical behavior of samples. Meanwhile, TiO_2_ was used as an active material to prepare electrodes for use as a high-performance supercapacitor.

## 2. Materials and Method

### 2.1. Materials

Deionized water was used to produce all products. The copper nitrate (Cu(NO_3_)_2_·3H_2_O), ammonium persulfate, aniline, titanium (IV) oxide (TiO_2_) (≥99.98%), poly(vinylidene fluoride) (CH_2_CF_2_)_n_) ammonia solution (NH_4_OH), N-methyl-2-pyrrolidone (C_5_H_9_NO), and hydrochloric acid (HCl) were procured from Sigma-Aldrich Chemical (Madrid, Spain). The sodium hydroxide (NaOH) and ethanol (C_2_H_5_OH) were purchased from Merck (Milan, Italy).

### 2.2. Fabricating TiO_2_-CuO Binary Composite

The TiO_2_-CuO composite was prepared as reported by Nguyen et al. [26]. 1 mmol of TiO_2_ was dispersed in 25 mL of absolute C_2_H_5_OH at 25 °C. Then, 25 mL of Cu(NO_3_)_2_ (0.02 M) were added to the solution suspension and stirred continuously for approximately 2 h at 500 rpm. Next, 0.25 M NaOH was added dropwise to this solution in order to adjust the pH to 10. After stirring for another hour, the resulting black TiO_2_–CuO precipitate was washed with deionized water and C_2_H_5_OH and the composite was dried at 70 °C in air for 5 h, and calcined at 400 °C for 3 h in air.

### 2.3. Synthesis of PANI@TiO_2_-CuO Ternary Composite

Two solutions were prepared. Solution-(A): 1.16 g of TiO_2_–CuO and 4.65 g of aniline were added into the 250 mL flask containing (1 M) HCl and stirred under ultrasonic for at least 1 h. Solution-(B): An amount of 11.50 g of ammonium persulfate was dissolved in 200 mL of 1 M HCl. Solution-(B) was then added to the aforementioned solution-(A), drop by drop, under constant magnetic stirring for another 24 h at a low temperature, allowing the aniline monomer polymerization and the obtainment of PANI@TiO_2_–CuO sample. The PANI was prepared under the same conditions but without TiO_2_–CuO, and also PANI@TiO_2_ with the presence of the TiO_2_ only [11,13,27].

### 2.4. Measurements

The chemical structure was characterized by FTIR (Bruker Instruments, Karlsruhe, Germany). The surface morphology of samples was characterized using SEM (JEOL JSM-5400, JEOL Ltd., Tokyo, Japan). The morphology of samples was further investigated by transmission electron microscopy (TEM) (JEOL-JEM 2100, JEOL Ltd., Tokyo, Japan). The surface composition was examined by X-ray photoelectron spectroscopy (XPS) analyses, utilizing a monochromatic 150 W Al X-ray source (ThermoScientific, Kyoto, Japan). The products were subjected to thermogravimetric analysis (TGA) technique (Hitachi-STA7200, Tokyo, Japan) under an N_2_ atmosphere. The products were heated from 25 °C to 900 °C at a heating rate of 10 °C/min. The optical proprieties and concentration of materials in the solutions was determined by Hitachi-U3000 spectrophotometer (Hitachi High Technologies Corporation, Tokyo, Japan). The crystalline property was evaluated using a Bruker CCD-Apex diffractometer (Bruker Corporation, Madison, WI, USA) (scanning voltage: 40 kV, scanning current: 30 mA, scanning angle: 5~70°, scanning speed: 5° min^−1^). The surface areas (S_BET_) of the samples were determined by N_2_ sorption–desorption on a Micrometrics ASAP 2420 analyzer (Micrometrics, Norcross, GA, USA) at 77 K using the Brunauer-Emmett-Teller method. Micropore volume and micropore surface area were estimated by the t-plot technique. The total pore volume (V_total_) of pores was determined from the N_2_ isotherms at $\frac{p}{p^{0}}=0.99$.

### 2.5. Electrochemical Investigation

The electrochemical properties of the product electrodes were tested by cyclic voltammetry (CV) using a conventional three electrodes electrochemical system using an eDAQ Model EA163 potentiostat [27]. Moreover, the electrochemical measurements were carried out by performing galvanostatic charge/discharge (GCD) analyses. The working electrodes were produced via coating of stainless steel (SS) current collectors with homogenous slurry of PANI@TiO_2_–CuO (80%) with carbon black (15 wt%) and CH_2_CF_2_)_n_ (5 wt%) in C_5_H_9_NO. The synthetized electrodes were dried in a vacuum oven at 70 °C for one night.

## 3. Results

### 3.1. Structure and Morphology Analysis

The XRD patterns of samples are investigated (Figure 1a). The XRD pattern of PANI exhibits the XRD peaks at 2*θ* = 20.2°, 25.03° and 28.69° [13]. These diffraction patterns proved the presence of crystalline domains in the amorphous PANI structure. The XRD patterns of crystalline CuO have characteristic diffraction peaks at 2*θ* values of 32.76°, 35.66°, 38.93°, 48.79°, 53.67°, 58.26°, 61.62°, 66.59°, and 68.22°, corresponding to (110), (002), (111), (202), (020), (202), (113), (311), and (220) lattice planes, respectively (JCPDS: 48-1548). Moreover, the characteristic peaks of TiO_2_ appear at 25.3°, 36.88°, 37.78°, 38.61°, 48.04°, 53.92°, 55.05°, 62.70°, and 68.85°, which can be attributed to the (101), (103), (004), (112), (200), (105), (211), (204), and (110) crystal planes, respectively [17,27]. Furthermore, the peaks diffraction pattern of the TiO_2_–CuO are a combination of signals from both TiO_2_ (JCPDS: 89-4921) and CuO (JCPDS: 48-1548), showing the crystal morphology and successful preparation of the TiO_2_–CuO structure. The XRD patterns of PANI@TiO_2_–CuO have the feature peaks of the PANI, as well as some peaks associated with TiO_2_–CuO. This proves the introduction of crystallinity to the PANI matrix and validates the preparation of hybrid material. From the characteristic diffraction, it appears that the broadness of XRD peaks of PANI is reduced and moved to higher 2*θ* values with respect to pure PANI. For instance, the peaks at 20.21° and 24.85° are slightly shifted to 20.62° and 24.97° for PANI@TiO_2_–CuO, respectively. This is attributed to the stronger coordination binding between the TiO_2_–CuO composite and the PANI matrix.

The FTIR spectra of samples are shown in Figure 1b. As seen in the FTIR spectra, the TiO_2_ presented signals around 3439 cm^−1^ assigned to angular deformation of adsorbed water and TiO_2_ structure with a strong band between 543 cm^−1^ and 680 cm^−1^ [27]. In addition, the FTIR spectra of TiO_2_–CuO exhibit three vibrations occurring at 524 cm^−1^, 605 cm^−1^, and 679 cm^−1^ for this sample, which can be attributed to the vibrations of Cu-O with TiO_2_, confirming the formation of highly TiO_2_–CuO composite. Moreover, PANI shows typical peaks at 1586 cm^−1^, 1500 cm^−1^, 1375 cm^−1^, 1312 cm^−1^, 1162 cm^−1^ and 826 cm^−1^ related to the stretching of the nitrogen quinone (Q) structure, benzene ring (B) structure vibration, C–N^+^ vibration, C–N vibration, in-plane C–H vibration, and out-of-plane C–H bending vibrations, respectively [28]. Compared with the FTIR spectra of hybrid material, the main change in the FTIR spectra of PANI@TiO_2_–CuO is the apparition of all bands of PANI with two new bands at 682 cm^−1^ and 522 cm^−1,^ that confirms the incorporation of polymer matrix in TiO_2_–CuO composite.

According to the XPS analysis (Figure 2a), the elements had been changed during the synthesis process, and compared with PANI, the XPS spectrum of PANI@TiO_2_–CuO new peaks Ti2p, Ti2s, Cu2p, and Cu3p related to TiO_2_–CuO composite. Moreover, the O/C ratio of PANI was lower than that of PANI@TiO_2_–CuO, proving the presence of bi-metal oxide in the PANI matrix.

Furthermore, as can be seen in Figure 3a, the high spectral resolution of the C1s peak for PANI showed three sub-peaks at 284.26 eV, 285.66 eV and 286.09 eV, which were assigned to the C–C/C–H, C–N and C=N bonds, respectively. The high-resolution spectra of ternary composite for N1s (Figure 3b) displayed two sub-peaks at 398.97 eV and 400.28 eV related to –N= and –NH–, respectively [29,30]. However, after the loading of polymer matrix onto the surface of TiO_2_–CuO, the C1s peak of PANI@TiO_2_–CuO showed a negative/positive shift in the binding energy compared to PANI; this spectrum (Figure 3c) reveals the existence of four peaks, namely 284.68 eV (C–C and C–H), 285.60 eV (C–N), 286.14 eV (C=N) and 287.19 eV (C=O). Further, the three peaks delimited in the N1s spectrum of PANI (Figure 3d) are considered as –N= (398.39 eV), –NH– (399.53 eV) and the –NH^+^ = peak at 400.35 eV, respectively [27,31]. Thus, these results demonstrate that polymers were successfully formed on the TiO_2_–CuO surface.

The N_2_ adsorption—desorption isotherms for PANI, TiO_2_–CuO and PANI@TiO_2_–CuO are presented in Figure 2b. It can be observed that all of the samples presented isotherms that are typical of mesoporous materials [32]. BET surface area (S_BET_) and total pore volume (V_total_) for samples are listed in Table 1. From the data, it can be inferred that S_BET_ and V_total_ of TiO_2_–CuO decreased after the PANI matrix formation on the TiO_2_–CuO surface, indicating that the polymer was created on the TiO_2_–CuO pore structure; similar behavior was also observed by Toumi et al. [33]. Moreover, PANI@TiO_2_–CuO presented a higher surface area than pure PANI.

SEM was adopted to examine the microscopic morphology of as-prepared samples. The TiO_2_ nanoparticles (Figure 4a) show some porosity with a dense agglomeration of nanoparticles, and were irregular in shape morphology, as previously reported in the literature [34]. However, as displayed in Figure 4b, the TiO_2_–CuO possesses a particular porous, netlike structure, with pores being well distributed, with an average pore-size of 20–30 microns. Meanwhile, the PANI (Figure 4c) is more irregular, possessing different shapes and sizes of agglomerates. Furthermore, it can be clearly observed that the PANI with a sphere structure and the mean void size of 20–30 nanometers are loaded on the TiO_2_–CuO surfaces without agglomerates in Figure 4c.

The samples are investigated using TEM and the images can be seen in Figure 5. The morphology of TiO_2_ particles is a spherical shape with regular shape and a size in the scale of 45–50 nm. As displayed in Figure 5b, the CuO spheres are uniformly distributed on the TiO_2_, and the inside is brighter than the outer layer’s black spheres, confirming that the formed TiO_2_–CuO composite has a unique hollow structure. Nevertheless, Figure 5c shows the image of the PANI, which is formed by nanofibers with a diameter of 15–20 nm. Van der Waal’s interaction caused the PANI structure to assemble into bundles. Further, the TEM image of hybrid material (Figure 5d) displays the uniform distribution of TiO_2_–CuO in the PANI matrix, due to strong interfacial adhesion between both. The average diameter of PANI@TiO_2_–CuO was found to be approximately 55–60 nm.

Figure 6a illustrates that the UV-Vis spectra of PANI and PANI@TiO_2_–CuO. PANI has two characteristic absorption peaks; a strong band at 333 nm corresponds to $\pi –\pi^{*}$ transition of the phenyl group and is attributed to the extent of conjugation along the PANI backbone. The band around 623 nm is due to the $n–\pi^{*}$ transition of the HOMO with LUMO orbitals of the (B) ring and (Q) rings, respectively [35,36]. For hybrid material, the shape of the spectra is similar to pure PANI, but there are modifications in the intensity and absorption edge due to the efficient interaction between nanosized TiO_2_–CuO and the PANI matrix. Moreover, the successful reinforcement of PANI with TiO_2_–CuO is ensured by the move in absorption edges, which can be attributed to $\pi –\pi^{*}$ and $n–\pi^{*}$ electronic transitions to lower (hypo-chromic) and higher (batho-chromic) wavelengths, respectively.

The minimum energy level needed for electrons to excite from a low-energy conduction band to a high-energy valance band is the optical band gap energy (*E_g_*). The *E_g_* values related with PANI and PANI@TiO_2_–CuO are calculated using Tauc’s equation [37]:(1)$${\left(ah\nu \right)}^{n}=B\left(h\nu -E_{g}\right)$$
where h is Planck’s constant, B is the absorption constant, α is the absorption coefficient, and ν is the frequency of light; n is a coefficient associated with the electronic transition and can take values of (½ and 2) if the transition is direct or indirect, respectively. The straight line’s intercept with the photon energy axis gives the *E**_g_* value of the sample and the obtained values are summarized in Figure 6b. The *E**_g_* value for PANI is 2.60 eV and that of PANI@TiO_2_–CuO is 2.27 eV, which can be ascribed to a newly formed intermediate energy level, through which facile excitation of electrons occurs [24,25].

The TGA results revealed that hybrid material shows different thermal behaviors than TiO_2_–CuO or pure PANI. The thermal decomposition of samples occurred in various steps, as shown in Figure 7. The first step has a mass loss of 3.32% for PANI@TiO_2_–CuO and 6.57% for PANI, which corresponds to the release of water molecules. The following steps of thermal decomposition were observed between 160 °C and 580 °C, and are associated with thermal decomposition of organic material. This result confirms the incorporation of the PANI matrix onto the TiO_2_–CuO surface. Furthermore, materials decompose gradually, and a sharp weight drop takes place between 600–900 °C. This illustrates that PANI@TiO_2_–CuO is more stable than PANI, but less stable than TiO_2_–CuO, indicating its potential application at higher temperatures.

### 3.2. Electrochemical Properties

The comparative CV of product electrodes in 1 M HCl electrolyte at a scan rate of 50 mV s^−1^, in the potential window of 0.0 to 1.0 V, are shown in Figure 8. The mass charge of the active samples on the material electrode is around 1 mg cm^−2^. For the PANI sample, the first redox peak at (0.46/0.28 V) relates to the Leucoemeraldine (LE) transition to the partly oxidized Emeraldine (E), resulting in a potential peak separation (∆*E_p_*) close to 180 mV; and the second redox peak (0.89/0.81 V) can be attributed to the (E) transition to the entirely oxidized Pernigraniline (PA) form that presents a ∆*E_p_* value of 80 mV [27]. Moreover, the CV of the PANI@TiO_2_–CuO electrode also shows two pairs of clear redox couples that conform to the pseudocapacitive behavior of PANI. These redox pairs reveal the redox transition between (LE) and (E), and between (E) and (PA). The oxidization peak at approximately 0.38 V is due to the transformation of the (LE) to conductive (E) state. The oxidization peak at 0.67 V can be assigned to the transformation of the (E) oxidation state to a fully oxidized (PA) form. The reduction peaks at 0.50 V and 0.31 V are due to the transformation of the (E) and (LE), respectively. As anticipated, the PANI@TiO_2_–CuO shows an enhanced noticeable electrical response when compared to PANI@TiO_2_ and pure PANI samples, which corresponds to the synergistic impact of TiO_2_–CuO and PANI as conductive fillers [38]. Meanwhile, the redox current density of this sample becomes larger and larger as the scan rate increases, indicating a good rate capability of the electrode. In other words, the formation of the PANI matrix on TiO_2_–CuO causes a shift in the redox peak values toward more negative potentials compared with PANI, and this phenomenon is due to the synergetic effect of the combined contributions from both inorganic and polymer [39]. Thereby, this pairs’ redox gives ∆*E_p_* values of 70 mV and 170 mV, respectively.

Additionally, Figure 9 shows CV performance of the PANI@TiO_2_–CuO, which is recorded in an electrochemical window from 0.0 V to 1.0 V at various scan rates of 5 mV s^−1^, 10 mV s^−1^, 20 mV s^−1^, 30 mV s^−1^, and 50 mV s^−1^. The CV analysis is beneficial to further understanding the electrochemical behavior and stability during electrochemical measurement processes [40]. The area of close curves gets larger with the increase of scan rate, while the shape of CV curves exhibit Faradic redox pairs, suggesting typical pseudo capacitance and double-layer capacitive behaviors of this supercapacitor electrode.

### 3.3. Galvanostatic Charge Discharge (GCD)

The charge storage properties were studied in more depth by GCD measurements. The GCD curves of the PANI@TiO_2_–CuO electrode display nonlinear shape with two voltage phases in the discharge operation, and also clearly indicate electrochemical behavior at various current densities (Figure 10), showing a perfect reversibility and effective charge/ion transport of the supercapacitor [40]. At relatively smaller current densities of 1.0 mA cm^−2^, the previous stage (1.0–0.5 V), with a comparatively short discharging time, corresponds to the electrochemical double layer capacitors (EDLC), while the last stage (0.5–0.0 V), with a longer discharge time, can be ascribed to an integration of EDLC and pseudo capacitor capacitance. The supercapacitor prepared using PANI@TiO_2_–CuO shows areal specific capacitances of 296.7 ± 0.2, 264.1 ± 0.2, 154.6 ± 0.2, 94.7 ± 0.2, and 82.8 ± 0.2 mF cm^−2^ at 1.0, 2.0, 5.0, 8.0, and 10 mA cm^−2^, respectively. It is possible that a current density above 5 mA cm^−2^ leads to irreversible processes [40]. Moreover, below 2 mA cm^−2^ leads to an improvement in the electrode performance. A typical comparison of GCD profiles of the PANI@TiO_2_ and PANI electrodes is displayed in the inset of Figure 11. Consequently, PANI@TiO_2_–CuO at a current density of 1.0 mA cm^−2^ displays the largest areal specific capacitance (296.7 ± 0.1 mF cm^−2^), which is better than those of PANI@TiO_2_ (182.5 ± 0.1 mF cm^−2^), and PANI (195.9 ± 0.1 mF cm^−2^). Moreover, the perfect conjugated structure of PANI backbone was destroyed and the conductivity of PANI@TiO_2_ decreased, due to presence of TiO_2_ on polymer matrix. A low conductivity causes slow electron transmission for the redox reaction of product, and leads to a low electroactivity of PANI@TiO_2,_ which eventually resulted in a low specific capacitance.

To evaluate and test the workable application of electrode materials, the long-term cycling stability at a current density of 1.0 mA cm^−2^ after 1500 cycles under continual GCD process is shown in Figure 11. The results suggest that the cycling stability of PANI@TiO_2_–CuO electrodes have a capacitance retention of 87.5 ± 0.1%, and it is better than that of other electrodes. This is fundamentally due to the PANI matrix being focused on the axial directions of the TiO_2_–CuO and being developed uniformly, with strong interfacial adhesion [41]. Conversely, PANI and PANI@TiO_2_ showed lower capacitance retention of 79.3 ± 0.1% and 84.7 ± 0.1%, respectively.

## 4. Conclusions

The new PANI@TiO_2_–CuO ternary composite was prepared via in-situ oxidative polymerization, while the TiO_2_–CuO was synthesized by a process involving the preparation of CuO in the presence of TiO_2_ by a facile method. The XRD technique results confirm the successful formation of TiO_2_–CuO and its presence in the emeraldine phase of the PANI matrix, and the crystallinity increases with the presence of TiO_2_–CuO in the ternary composite structure. The structure, morphology, and thermal properties of materials were systematically characterized using FTIR, XPS, XRD, SEM, TEM, UV-Vis, and TGA. The results indicated that TiO_2_–CuO was dispersed uniformly in the PANI matrix. Due to such dispersion of the TiO_2_–CuO composite, the PANI@TiO_2_–CuO ternary composite exhibits dramatic improvements compared to the neat PANI. The surface area and pore structure properties of the materials under investigation are measured by BET. Moreover, the PANI@TiO_2_–CuO material shows a highly enhanced capacitance in comparison with PANI and PANI@TiO_2_. This superior capacitance results from the combination of electrochemical behaviors of PANI and TiO_2_–CuO. The ternary composite shows long-term cycling stability without any noticeable degradation for over 1500 cycles. These findings provide a facile strategy to prepare hybrid materials of PANI and TiO_2_–CuO with a highly improved capacitance, which we believe will have possible applications in the future.

## Figures and Tables

**Figure 1 polymers-14-04562-f001:**
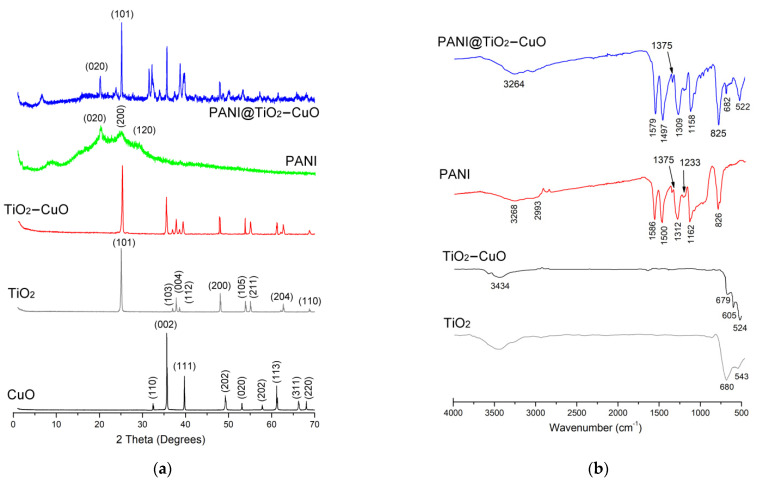
(**a**): XRD patterns and (**b**): FTIR analysis of samples.

**Figure 2 polymers-14-04562-f002:**
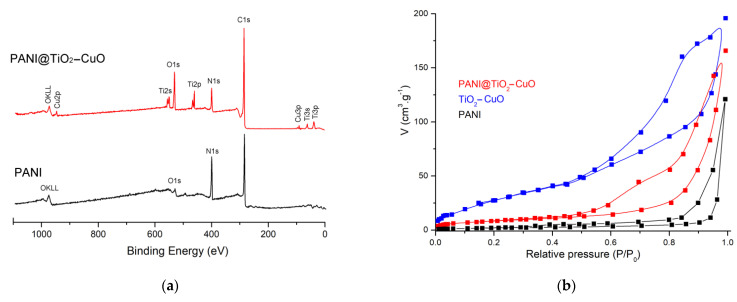
(**a**): Survey XPS spectra and (**b**): N_2_ sorption isotherms of samples.

**Figure 3 polymers-14-04562-f003:**
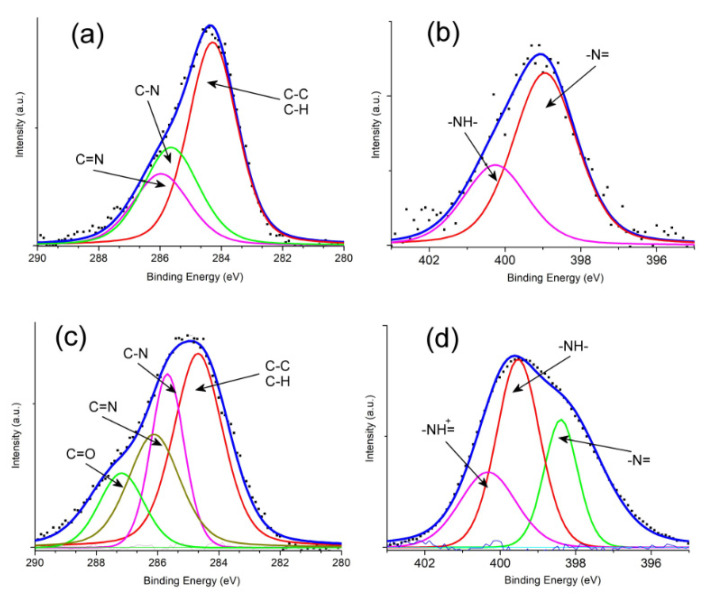
XPS spectral signals: (**a**) C1s of PANI, (**b**) C1s of PANI@TiO_2_–CuO, (**c**) N1s of PANI@TiO_2_–CuO, and (**d**) N1s of PANI.

**Figure 4 polymers-14-04562-f004:**
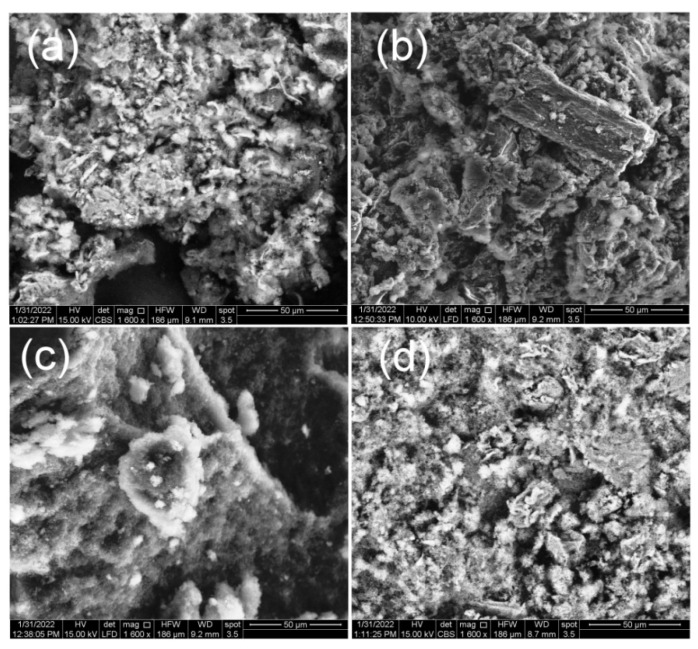
SEM images of: (**a**) TiO_2_, (**b**) TiO_2_–CuO, (**c**) PANI, and (**d**) PANI@TiO_2_–CuO.

**Figure 5 polymers-14-04562-f005:**
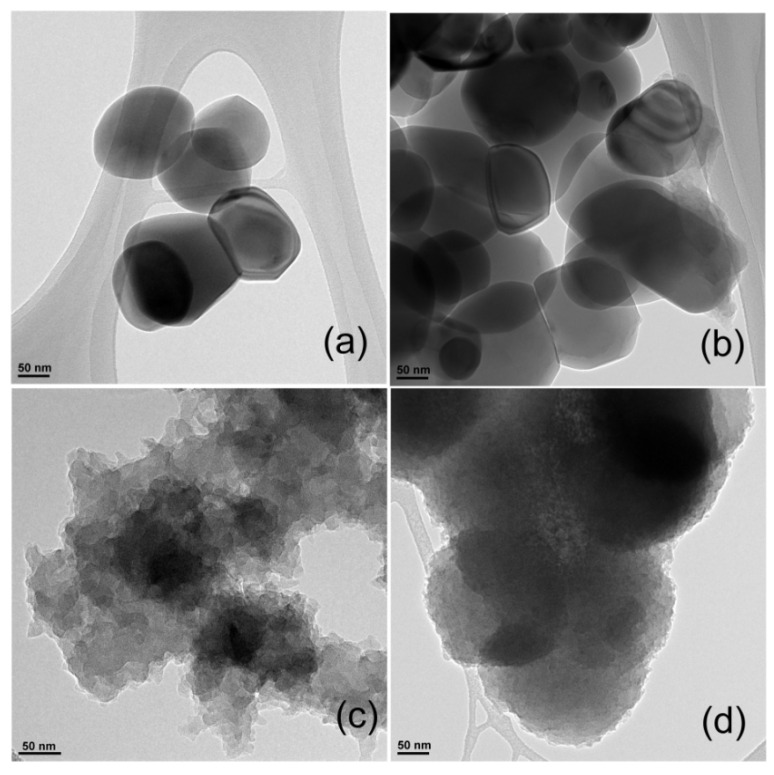
TEM images of: (**a**) TiO_2_, (**b**) TiO_2_–CuO, (**c**) PANI, and (**d**) PANI@TiO_2_–CuO.

**Figure 6 polymers-14-04562-f006:**
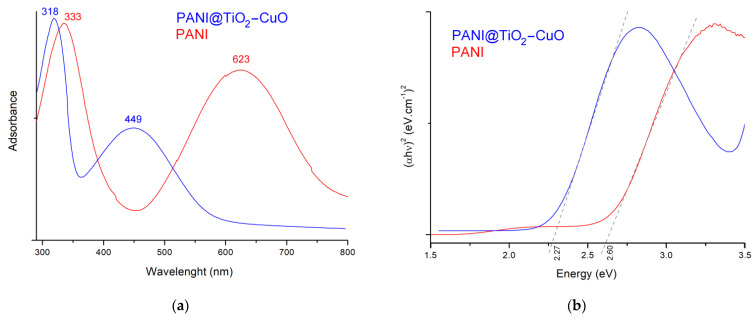
(**a**): UV-vis spectroscopy absorption spectra and (**b**): Tauc plots of samples.

**Figure 7 polymers-14-04562-f007:**
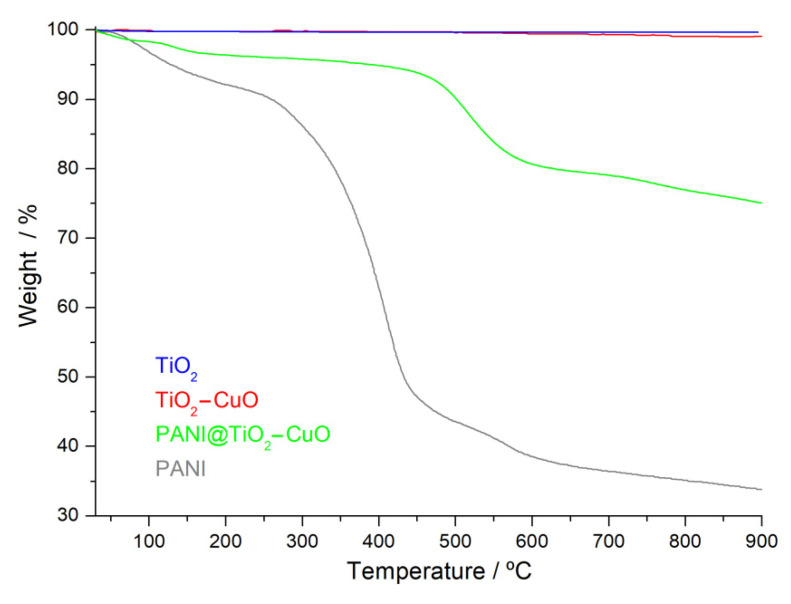
TGA curves of samples.

**Figure 8 polymers-14-04562-f008:**
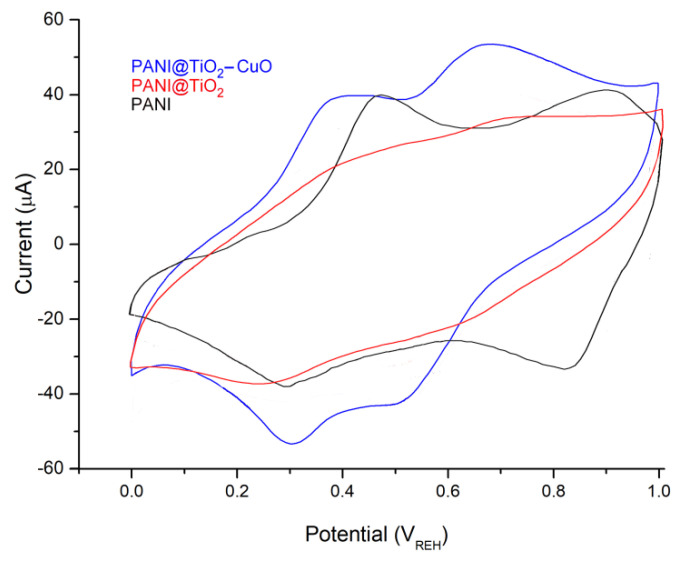
Cyclic voltammograms recorded for a stainless in HCl solution (1M) scan rate 50 mV s^−1^.

**Figure 9 polymers-14-04562-f009:**
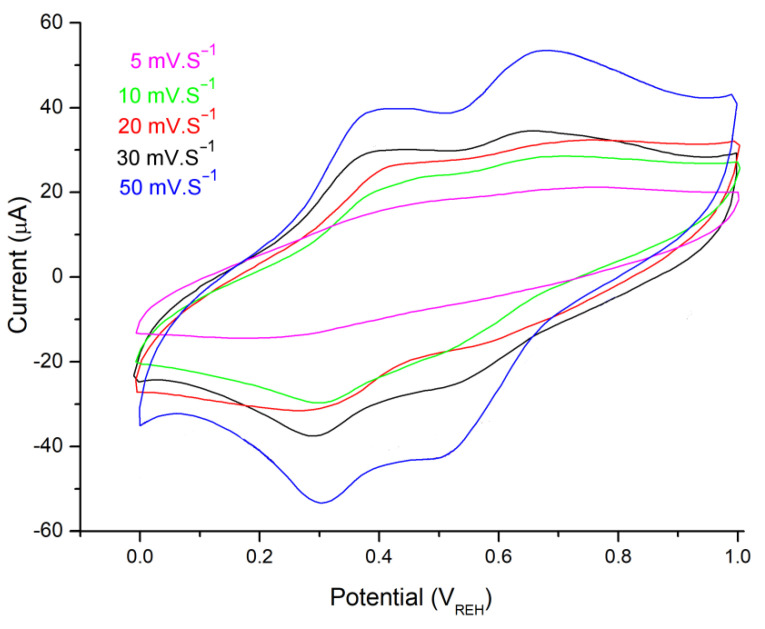
CV profiles of the PANI@TiO_2_–CuO electrode at various sweep rates at 1.0 A g^−1^ in HCl solution (1 M).

**Figure 10 polymers-14-04562-f010:**
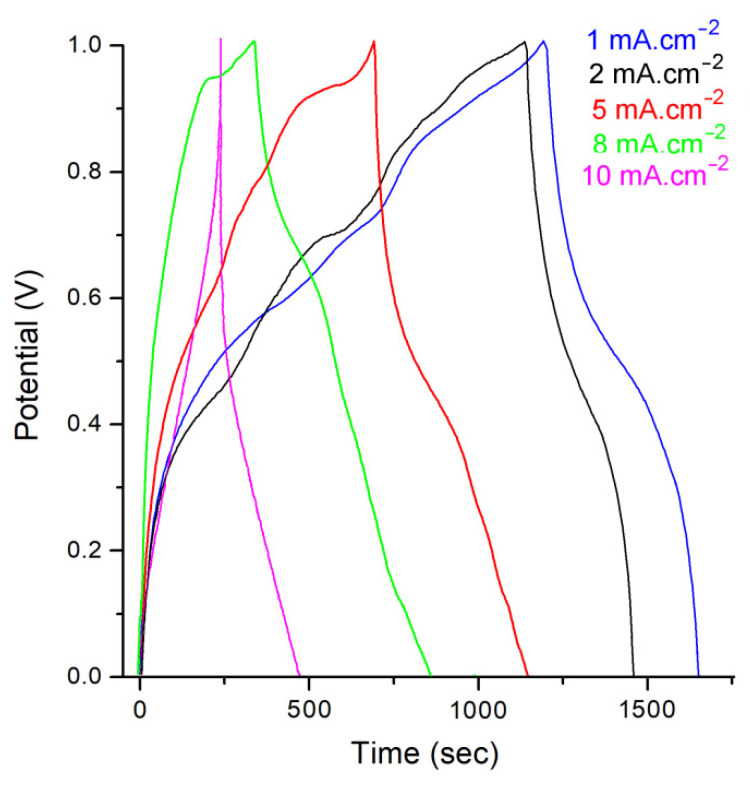
Galvanostatic charge–discharge (GCD) curves of the PANI@TiO_2_–CuO electrode at different current densities.

**Figure 11 polymers-14-04562-f011:**
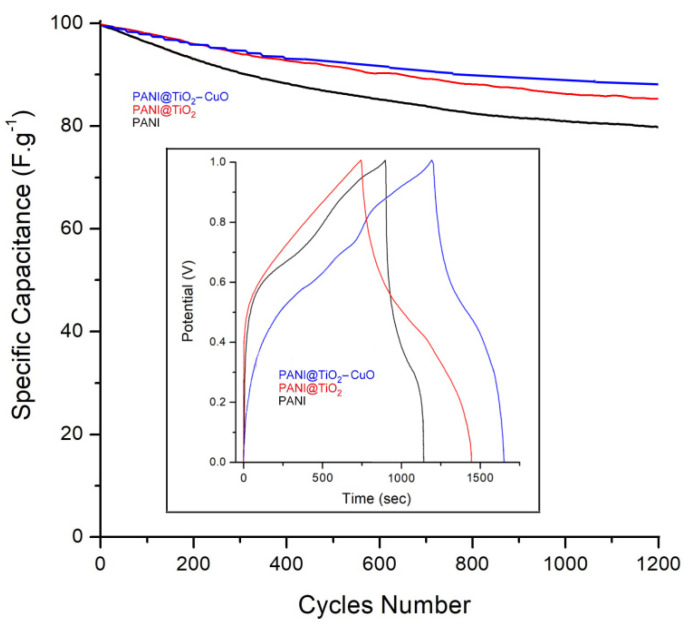
The cycling performance of PANI, PANI@TiO_2_ and PANI@TiO_2_–CuO electrodes at a current density of 1 A g^−1^; The inset demonstrates the comparative GCD curves of PANI, PANI@TiO_2_ and PANI@TiO_2_–CuO at a current density of 1.0 mA cm^2^.

**Table 1 polymers-14-04562-t001:** Materials’ textural characteristics obtained from N_2_ adsorption-desorption data.

Material	PANI	PANI@TiO_2_	TiO_2_-CuO	PANI@TiO_2_-CuO
S_BET_/(m^2^ g^−1^)	25 ± 0.5	53 ± 0.5	132 ± 0.5	108 ± 0.5
*V_DR_* (N_2_)/(cm^3^ g^−1^)	1.50 ± 0.01	1.32 ± 0.01	1.98 ± 0.01	1.94 ± 0.01
V_meso_/(cm^3^ g^−1^)	0.01	0.02	0.05	0.04
V_macro_/(cm^3^ g^−1^)	0.01	0.08	0.14	0.11
V_tot_ pore/(cm^3^ g^−1^)	0.02	0.10	0.19	0.15

## Data Availability

Not applicable.

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
