# Peer review of "Facile Synthesis and Electrochemical Characterization of Polyaniline@TiO2-CuO Ternary Composite as Electrodes for Supercapacitor Applications"

_polymers, 2022, doi:10.3390/polym14214562_

Round 1

Reviewer 1 Report

This paper reported the synthesis and electrochemical characterization of poly aniline@TiO2-CuO ternary composite as electrodes for supercapacitor applications. Although the structural and electrochemical properties were analyzed in detail, the work is not suitable for publication in a Polymer journal in this form. Therefore, I recommend a major revision.

Abstract section:

-           Line (30): "new ternary composite electrode." Clarify the meaning in the manuscript.

-           The electrochemical behavior (stability test) tested over 1500 cycles did not make sense. The 5000th cycles are required, at least, for supercapacitor application.

Introduction section:

-           Line (50): "However, to improve environmental stability, CPs was added to the metal oxides (MO) nanoparticles". More clarification is required for this sentence.

-           Line (54): "CPs@MO nanomaterials have fascinated investigators due to their excellent properties, such as their excellent physical and chemical features and lower-tier toxicity". This sentence needs to be re-written to show the properties (features).

-           Line (59): "Investigators have been studying the structural, electrical, and optical features of PANI-based TiO2 nanomaterials as an electrode for supercapacitor". Comparisons should be provided with recent studies to show the novelty of this work.

Materials and method section:

-           The source of TiO2 is missing.

Results section:

-           Line (142): Please check the (JCPDS: 13-0835 and JCPDS: 65-7994) card numbers are not related to CuO.

-           Line (195): "from the data, it can be inferred that SBET and Vtotal of TiO2–CuO decreased after the PANI matrix formation". The data values for TiO2–CuO are missing in the table.

-           The figure capture is unclear in some figures (Figure 7b and Figure 8). Also, in line (288): "Figure 12 shows CV performance of the PANI@TiO2–CuO". Please correct the figure no.

-           Line (297): "shape with two voltage phases in the discharge operation and also clearly indicate electrochemical behavior at various current densities". Please check the Figure.

-           Line (304): "Besides, that supercapacitor prepared using PANI@TiO2–CuO shows areal specific capacitances of 296.7, 264.1, 154.6, 94.7 and 82.8 mF.cm−2" why the electrode shows very low-rate capability (~ 28%)?

-           Line (306): "A typical comparison of GCD profiles of the PANI@TiO2 and PANI electrodes is displayed in Figure 9-a". The GCD profiles of the PANI@TiO2 and PANI electrodes are missing in the manuscript.

-           "PANI@TiO2-CuO displays the largest areal specific capacitance (296.7 mF.cm−2), which is better than those of PANI@TiO2 (182.5 mF.cm−2), and PANI (195.9 mF.cm−2) ". Why PANI@TiO2 showed low capacitance than PANI? Please explain this in the text.

-           Line (318): "PANI@TiO2-CuO depicts a steadily continual GCD operation for 1,500 cycles without a remarkable degradation and with capacitance retention of 87.5%". Why is capacitance retention decreased to 87.5% if no remarkable degradation exists?

Author Response

Dear Reviewers,

The authors would like to extend great thanks to you for your time and valuable comments and suggestions, which help us improve our paper to a more scientific level.

We have revised our manuscript carefully, and quite a lot of changes have taken place.

The detailed explanations are listed point-by-point as follows.

Yours sincerely.

Reviewer 2 Report

The manuscript deals with the synthesis and characterizarion of polyaniline@TiO2-CuO ternary compositea as well as its possible practical application as supercapacitor. The topic is of interest. The composite material synthetized in current work is another one example of combination TiO2 and CuO nanoparticles and polyaniline. Although, the application is diferent.

The experiment is logically designed. Methods are described well. Conclusions are supported with data ontained.

Nevertheless, the manuscript needs significant revision.

1. English needs to be checked and revised. There are many badly constructed phrases throughout the text. For example,

Abstract, "This research reports the facile, low-cost, controlled and evaluation of properties...",

Lines 23-23, "The structure, morphology, and thermal properties of... were systematically characterized using FTIR spectroscopy...... X-ray diffraction (XRD) analysis, X-ray photoelectron spectroscopy (XPS) analyses, transmission electron microscope (TEM), scanning electron microscopy (SEM), thermogravimetric analysis (TGA), and BET surface areas"

and so on.

2. Abbreviations need revision. Many of them are used just one-two times throughout the manuscrupt and have to be removed.

3. The majority of experimental data (Table 1, specific capacitance values at lines 305-606 and 308-309, capacitance retention values at line 323) are based on the single measurement results making them unreliable. How many parallel measurements were carried out? The average value with corresponding SD is required for the each numerical parameter.

4. Fig. 7b and 8a, cyclic voltammogram of the supporting electrolyte is needed.

5. Line 288, the reference to Figure 12 in mentioned instead of Figure 8a.

6. All variables throughout the manuscript should be presented in italics.

7. Section 2.1, the names of chemicals are written from the capital letter inside the senstences. Please, replace capital letters with the small letters.

8. Fig. 7 and Fig. 8 have to be rearranges as: Fig. 7 should contain TGA curves only, i.e., remove part 7b, which has to be added to Fig. 8 as part 8a. Parts 8a and 8b in the current version of Fig. 8 have to become parts 8b and 8c, respectively, in revised version. It will be more logical and easier to understand.

Author Response

(The authors gave the same response as above.)

Round 2

Reviewer 1 Report

The manuscript was improved and can be published in Polymers Journal.

Author Response

Thank you very much for your comments that are helped me to rewrite this manuscript.

Reviewer 2 Report

The revised version is just partially improved vs. initial submission. The most important drawback (remark 3) is ignored. There are other points needed to be corrected.

1. English in the Abstract is corrected but still insufficient. I would suggest the following corections (changes are marked in bold)

a) "This research reports the facile, controlled, low-cost and evaluation fabrication of properties of polyaniline matrix deposited on Titanium dioxide and Copper(II) oxide ternary-composite (PANI@TiO2CuO)-based electrode material for supercapacitor application."

replace with "This research reports the facile, controlled, low-cost fabrication and evaluation of properties of polyaniline matrix deposited on titanium dioxide and copper(II) oxide ternary-composite (PANI@TiO2CuO)-based electrode material for supercapacitor application."

b) "The structural and physical properties were evaluated based on the results of FTIR spectroscopy, X-ray diffraction (XRD) analysis, X-ray photoelectron spectroscopy (XPS) analyses, transmission electron microscope (TEM), scanning electron microscopy (SEM), thermogravimetric analysis (TGA) and BET surface areas analysis."

replace with "The structural and physical properties were evaluated based on the results of FTIR spectroscopy, X-ray diffraction (XRD) analysis, X-ray photoelectron spectroscopy (XPS), transmission electron (TEM) and scanning electron (SEM) microscopy, thermogravimetric analysis (TGA), and BET surface areas analysis."

2. Abbreviations used 1-3 times were kept in the manuscript. These are DW, APS, ANI, PVDF, NMP.

3. Remark 3 to the initial version of the manuscript is not taken into account. The answer regarding number of parallel measurements is not enough. The average value does not allow evaluate the significance in the data difference. Corresponding changes have to be inserted i.e. the "average value ± SD" is required for the each numerical parameter. These changes to be inserted for data in Table 1, specific capacitance values at lines 317-318 and 323-324, capacitance retention values at line 333, which are shown as a single measurement results making them unreliable.

4. Figure 9 caption, remove the following text "at different current densities" which is valid for Figure 10 caption.

Author Response

We found the reviewer comments helpful and have incorporated the all of the changes suggested. A detailed list of our responses to the review comments and a version of the revised manuscript with changes marked-up using the yellow font color are included. 
We have made the best effort to provide very thorough responses to all of the issues raised by reviewers. We think we have sufficiently addressed the reviewers’ comments, and we hope you will find the revised manuscript acceptable for publication in Polymers.

Round 3

Reviewer 2 Report

This version of the manuscript can be accepted to publication.